



# Assessing and mitigating the radar - radar interference in the German C-band weather radar network

Michael Frech[1], Cornelius Hald [1], Maximilian Schaper [1], Bertram Lange [2], and Benjamin Rohrdantz [2]

[1]Deutscher Wetterdienst, Observatorium Hohenpeißenberg, Albin-Schwaiger-Weg 10, 82383 Hohenpeißenberg
[2]Deutscher Wetterdienst, Sasel

**Correspondence:** Michael Frech
Michael.Frech@dwd.de

**Abstract.** The national German weather radar network operates in C-band between 5.6 and 5.65 GHz. In a radar network, individual transmit frequencies have to be chosen such that radar-radar induced interferences are avoided. In an unique experiment the Hohenpeissenberg research radar and five operational systems from the radar network were used to characterize radar-radar induced interferences as a function of the radar frequency. The results allow to assess the possibility to add additional C-band radars with magnetron transmitters into the existing network. Based on the experiment, at least a 15 MHz separation of the nominal radar frequency is needed to avoid a radar-radar interference. The most efficient mitigation of radar-radar interference is achieved by the "Radar Tango". Latter refers to the synchronized scanning of all radar systems in the network. Based on those results, additional C-band radar systems can be added to the German weather radar network if a further improvement of the radar coverage is needed.

## 1 Introduction

The German weather radar network consists of 17 state-of-the-art dualpol weather radar systems that are operated at C-band in the frequency range between 5.6 and 5.65 GHz. We operate EEC's DWSR5001SDP/CE radar system, technical details about this system are given in Frech et al. (2017). The radar network is operated by the German Meteorological Service (Deutscher Wetterdienst, DWD). Weather radars are one of the most important data sources for a wide range of applications. They are the backbone for nowcasting products, which provide reliable warnings to the public in severe weather situations and they are the only data source to provide quantitative areal information on the precipitation amount and type, which is an essential information for hydrological and flood warning applications. Radar data are also assimilated into numerical weather prediction models. The DWD scan strategy has an update frequency of 5 minutes and consists of two basic scan types. For hydrological applications, a terrain following scan (referred to as the precipitation scan, PCP) is implemented, where the elevation angle is adjusted as a function of the azimuth angle based on the surrounding orography. This sweep is followed by a volume scan that samples the three-dimensional space at 10 elevations. The volume scan is operated in staggered PRF (pulse repetition frequency) which provides unambiguous Doppler velocities of $\pm 32$ ms$^{-1}$. The parameterizations of those scans are listed in Appendix A. The five minute sequence ends with a so-called birdbath scan, which is used to calibrate the differential reflectivity (Frech and Hubbert, 2020).





Before radar data can be used as input for operational products, extensive quality control is needed to identify and mitigate all non-meteorological contributions to the radar signal (Werner and Steinert, 2012).

    Radar measurements close to the surface are only available at a limited range relative to the radar because surface clutter can only be avoided if the radar beam (typically 1° in diameter) has a certain minimum elevation (typically 0.5 to 1°). This is of course dependent on site conditions and the surrounding orography. With a typical operating range of up to 180 km, the

curvature of the earth is relevant as well. The actual coverage with radar data up to 3 km above the surface of the network in Germany is shown in Figure 1. From this we can identify areas where coverage could be significantly improved, if additional radar systems are installed. When filling gaps, it is proposed to consider C-band systems. Staying with the same frequency band minimizes the modifications that are necessary in the radar processing stream down to the final radar product, because varying frequency dependent scattering characteristics are avoided. Yet, if additional C-band radar systems are added to the

network, radar-radar induced interferences may occur and have to be avoided if possible.

    DWD operates magnetron transmitters with a nominal pulse power of 500 kW. Magnetron transmitters are very common in radar networks all over the world, because they are affordable and relatively robust units compared to other available transmitters on the market. The radar design however has to account for some special characteristics of the magnetron tube. The transmit phase for each pulse is random, which means that transmitted pulses are non-coherent. Coherency can be achieved

with a reference oscillator. The actual transmit frequency depends on the duty-cycle of the magnetron and on temperature. The duty-cycle is dependent on the pulse width (PW) and the PRF. The DWD system is operated with a PW of 0.4 and 0.8 μs in a PRF range between 600 and 2410 Hz. With this operating range, the transmit frequency is constant within a range of ±0.5 MHz. Accounting for typical temperature variations on site (less than 2 K due to air-conditioning), the overall transmit frequency is within ±1 MHz with respect to the nominal transmit frequency of a magnetron. Weather radars in C-band operate

as primary users in the frequency band between 5600 MHz and 5650 MHz in Germany.

    Interferences from external sources that operate at C-band are frequently visible in day-to-day operations. Interferences compromise radar data quality and the problem is expected to increase further (Saltikov et al., 2016). The urgent need to secure and protect atmospheric remote sensing frequency bands is summarized in an overview by Palmer et al. (2021). Even though weather radars are the primary user in their band, WiFi devices at C-band are allowed to operate there as well if they use

dynamic frequency stepping (DFS). Ideally, DFS should recognize the weather radar in the vicinity and consequently adjust the WiFi transmit frequency to avoid an interference with the weather radar. This works well in theory, but in practice the national frequency authorities (BNetzA, Bundesnetzagentur in Germany) are busy to enforce the proper usage of WiFi devices. A WiFi interference is typically seen as line shaped structure in radar products. If the WiFi device is close to the radar, complete sectors of a radar sweep may be affected.

A realtime identification of a interference is implemented in the GAMIC signal processor. It is based on the evaluation of the standard deviation of the power (STD) for a batch of pulses within a ray (ray width is 1°), using I&Q-data. Typically, a batch consists of about 60 pulses, which varies depending on the PRF and the rotation speed of the antenna. Internally I&Q-data are sampled to 25 m bins ("raw rangebin"). After the signal processing, radar moments are averaged to 250 or 1000 m bins, the so-called processed rangebins. A large standard deviation is indicative of the presence of non-coherent power that can be





attributed to an external interference. Together with the so-called signal quality index (SQI), which quantifies the coherence of the received signal with the transmitted pulse, an interference contamination can be identified. We check for SQI < 0.6 and mean STD > 0.6 to identify an interference. This method is used operationally and proves to be very successful to locate and shut down interference sources by the frequency authority (see Schaper et al. 2022).

Contrary to line-shaped signatures from WiFi devices, a radar-radar interference is visible as a spiral shaped signature
in a PPI (plan position indicator) due to the moving antennas, pulsed waveform, and high peak signal power. This kind of interference can be attributed to a remote radar that operates close to or at the same radar frequency. These interferences were observed only rarely. As a means of avoiding a radar-radar interference, the scanning of all DWD radars is synchronized: All radars point to the same elevation and azimuth angle at the same time. This is denoted as the "Radar Tango" and has been implemented in summer 2017. All radars start their scanning at 0° in azimuth and 0.8° in elevation. Within a 5-minute
sequence, a synchronization of the pointing among the 17 radar systems to a precision of <2° is achieved. Qualitatively, this approach has significantly reduced the number of radar-radar interferences. However, a quantification is missing. The Radar Tango was easy to implement because the radar network is homogeneous in terms of hardware and software capabilities.

In this paper, we assess the performance of the "Radar Tango" and investigate whether it is possible to introduce additional C-band systems into the radar network. This will be evaluated in terms of the necessary frequency separation to avoid radar-
radar interference within the C-band network. Since we need specific design guide lines that work in operational condition with a 24/7 operation of radar systems, an experiment has been designed by using five operational radar systems and the Hohenpeißenberg (MHP) research radar of the German radar network. The allocated transmit frequencies of these operational radar systems cover the C-band frequency range and are therefore well suited for such an experiment. The radars are configured to point directly towards the research radar in order to generate a possible interference. The unique aspect of this setup is that
we carried out this dedicated testing over a span of about four weeks under quasi-operational conditions. The test was designed in such a way that the operational scanning was not interrupted. This allows for an assessment of the interference situation under a wide range of beam propagation conditions.

As part of the experiment, we also assess the suitability of the "Radar Tango" as a means of minimizing the radar-radar interference. This is achieved by comparing periods with synchronized and unsynchronized scanning of the Hohenpeißenberg
radar with the network radars (Chapter 3.3). The results of these investigations are summarized in the concluding section.

## 2   Setup of the interference experiment

The principle idea of the experiment is to detect and quantify the interference caused by other radars operating in the C-band at the research radar Hohenpeißenberg and to evaluate the influence of the relative difference in transmit frequencies. The MHP radar is located in Southern Germany. The radar is on a solitary mountain 1000 m above sea level, about 20 km north
of the Alps and 60 km South-West of the city of Munich. Its location and the locations of the five operational systems used in this study are shown in Figure 2. An interference situation is achieved by intentionally pointing each of the operational radar systems towards the MHP radar for about 30 seconds ("fixed scan radars"). During this time, a standard precipitation scan is




**Figure 1.** Radar data coverage up to a height of 2 km above the surface based on the precipitation scan, which is a terrain following scan. Shown is the coverage of the operational radar network consisting of 17 dualpol C-band radars.



**Table 1.** A summary of operational radar sites used in the interference test. Azimuth [°] / elevation [°] describe the angle configuration of the pointing scan towards MHP.

| Site | TX frequency (MHz) | distance to MHP (km) | azimuth [°] / elevation [°] |
|------|-------------------|---------------------|----------------------------|
| Türkheim (TUR) | 5600 | 126 | 133 / 0.5 |
| Feldberg (FBG) | 5670 | 225 | 90.9 / 0.5 |
| Memmingen (MEM) | 5630 | 65 | 114.1 / 0.5 |
| Eisberg (EIS) | 5640 | 219 | 208.4 / 0.5 |
| Isen (ISN) | 5625 | 91 | 243.4 / 0.5 |
| Hohenpeißenberg (MHP) | 5640 | - | - |

acquired at MHP which is then analyzed for an interference signal. All five operational radar systems are located within a range of 225 km from the MHP radar in Southern Germany (Figure 2). The radars operate at transmit frequencies between ±30 MHz

of the Hohenpeißenberg transmit frequency of 5640 MHz (see Table 1). With the permission of the regulation authorities, the Feldberg (FBG) radar is operating outside the official C-band in order to minimize interference with a neighboring French weather radar.

Because the operational scanning cannot be interrupted for this experiment, we had to clear approximately 30 seconds of scanning time at the end of each operational 5 minute cycle. To achieve this, it was sufficient to temporarily suspended one of

the two birdbath scans (for the two pulse widths).

In order to obtain conclusive results, only one radar at a time is transmitting towards the MHP radar. Using the operational radar software, the set of tests using the five operational systems can be executed in repeating 30 minute intervals (Table 2). At the same time, the MHP scheduler is identical to the operational scan, performing the same PCP and volume scan every 5 minutes. The only difference is the scanning in the 30 second time slot at the end of each 5 minute cycle. At MHP, a PCP-like

sweep is acquired. All scan configurations can be found in Appendix A.

This test setup was incorporated in the the operational scan cycle. In doing so, we were able to capture data with a range of meteorological conditions. Especially situations with anomalous propagation (stably stratified atmosphere), but also precipitation events were captured during the test period.

The 24/7 operation of the 30 minute scan sequence summarized in Table 2 was initiated on 2019-12-06 and ended on 2019-

12-20. During this time period, a number of systematic variations on the transmit and receiving end of the test were carried out. On the transmit side there are two periods with a single PRF of 1000 Hz and 600 Hz, respectively. Those periods are followed by a staggered PRF setting of 800/600 Hz which is realized with a time sampling (TS) of 50 pulses (50 pulses with PRF 800 Hz, followed by 50 pulses with a PRF of 600 Hz). On the receiving side, the precipitation scan has a PRF of 600 Hz, or a staggered PRF of 800/600 Hz. Additionally, the precipitation scan was operated in receive mode only for some time, where

the transmitter of the MHP radar was turned off. The configurations are summarized in Tables 3 and 4.



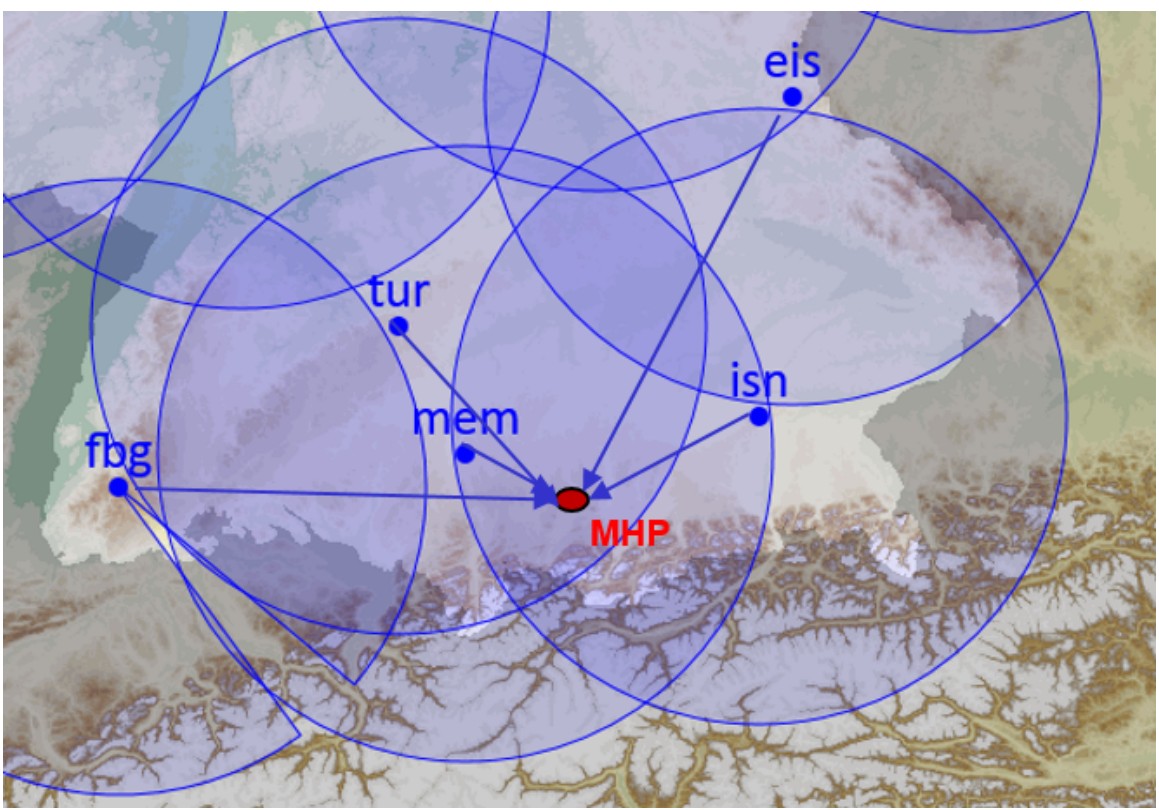

**Figure 2.** Location of the radars that are used in this test. Shown are in blue the operational systems, and in red the research radar MHP. Circles have a radius of 150 km. The arrows towards MHP symbolize the interference scan from operational radars.

**Table 2.** The repeating 30 minute scan sequence during the interference test. Precipitation scan and volume scan together take about 4:30 minutes to finish, and are repeated every 5 minutes.

| Time (mm:ss) | system → MHP |
|---|---|
| 04:30 - 05:00 | TUR |
| 09:30 - 10:00 | MEM |
| 14:30 - 15:00 | FBG |
| 19:30 - 20:00 | EIS |
| 24:30 - 25:00 | ISN |
| 29:30 - 30:00 | no dedicated interference |

It is known that the weather situation has a substantial influence on the propagation of electromagnetic waves in the atmosphere (e.g. Doviak and Zrnic (2006)). Whether a remote radar can be seen by the MHP radar depends on the static stratification of the atmosphere. This can be assessed by applying a simple yet effective approach for determining the position of a radar





**Table 3.** Transmit test configurations 2019-12-06 to 2019-12-20. Operational radars pointing towards radar MHP.

| date | interference scans |
| --- | --- |
| 2019-12-04 to 2019-12-09 | single PRF 1000 Hz |
| 2019-12-09 to 2019-12-16 | single PRF 600 Hz |
| 2019-12-16 to 2019-12-20 | staggered PRF 800/600 Hz |

beam, the effective earth radius or "4/3-model" (detailed information and for this and additional comparable models for differ-
ent atmospheric states can be found in Holleman and Huuskonen (2013)). The "4/3-model" represents a standard atmosphere
with a refractivity gradient $\frac{dN}{dh} = -42$ km$^{-1}$ (Martin and Vaclav, 2011).

With this, the height of the beam of a radar over any other point on earth in dependence on range and elevation angle can be
approximated:

$$H_b = \sqrt{r^2 + (k_e \cdot r_e)^2 + 2 \cdot r \cdot k_e \cdot r_e \cdot sin(\theta)} - k_e \cdot r_e \tag{1}$$

$H_b$ is the beam height over ground, $r$ is the range of the beam, $k_e$ is $\frac{4}{3}$ or the factor for the effective earth radius, $r_e$ is the earth's
radius at the respective latitude and $\theta$ is the elevation angle. The radar EIS will be used as an example for the calculation. The
radar EIS is at 49.539° N and 799 m above sea level. The elevation angle $\theta$ during the measurements was 0.5° and the distance
between the two radars is 219 km. The vertical size of the 3 dB beamwidth at EIS is 0.91°, so the (vertical) radius of the beam
can be described as:

$$r_{b,d} = (2 \cdot r \cdot \pi) \cdot \frac{w_b}{360} \cdot \frac{1}{2} \tag{2}$$

with $r_{b,d}$ being the radius of the beam at distance $d$ and $w_b$ is the beam width. This results in a beam radius of 1.739 km at
a distance of 219 km. With the "4/3-model", the lower 3 dB-edge of the EIS-beam over radar MHP (located at 47.793° N,
1000 m asl) is expected to be at 2.1 km. So under normal propagation conditions no interference may be expected. As an
estimate, EIS could be visible by MHP with an EIS elevation angle $\theta = 0°$ if $\frac{dN}{dh} = -150$ km$^{-1}$. This would be achieved
with a gradient of about 8 K/100 m. This is not uncommon during clear sky winter-time high pressure situations with strong
radiative cooling during the night. As will be shown later, frequent interferences are detected during the experiment which are
attributed to the radar EIS. So periods with strong vertical temperature gradients were present during the experiment .

## 3 Analysis

All MHP precipitation scans at the end of each 5 minute sequence are analyzed for interferences. If the condition STD > 0.6
and SQI < 0.6 is satisfied, a range bin is classified as disturbed by an external interference.

In a similar way, interferences are identified from the MHP precipitation scans that happen at the beginning of each five-
minute scan cycle as part of the operational scanning schedule. Those results are used to evaluate the "Radar Tango" in com-



**Table 4.** Radar MHP test configurations 2019-12-06 to 2010-12-20.

| date | interference scans |
|------|--------------------|
| 2019-12-04 to 2019-12-16 | single PRF 600 Hz |
| 2019-12-16 to 2019-12-17 | staggered PRF 800/600 Hz |
| 2019-12-17 to 2019-12-19 | staggered PRF 800/600 Hz, TX off |
| 2019-12-19 to 2019-12-20 | staggered PRF 800/600 Hz, TX on |

parison to the MHP precipitation scans that are either synchronized or not with the starting azimuth angle of the other radars. Those precipitation scans are available every 5 minutes.

### 3.1 Example for radar-radar interference

Before analyzing the results statistically for the different measurement scenarios, we first show some examples. The spiral shaped signature of radar-induced interference in PPIs can be nicely illustrated in B-Plots where the interference then appear as straight lines (Figures 3 and 4). Figure 3 shows the recorded uncorrected reflectivity (UZh) during two PCP scans within 5 minutes of each other. The upper image is from a scan where the transmitter at MHP was turned off during the scan, whereas the MHP transmitter was in an operational state in the lower image. In the latter case, clutter power is clearly visible close to the radar. Both images show distinct lines, reaching from the top left to the bottom right. They mark the interference from the EIS radar. The interferences appear at different ranges at the same time and the lines are parallel to each other. The distance between these lines is determined by the PRFs of the sending and receiving radar. It appears as if the strength of the interference increases with increasing distance from the radar, but this is an artifact of the range correction. In reality, the strength of the signal is constant in all ranges. Figure 4 shows the STD moment for the two scans described above. In most of the scanned area, STD is at ≈0.5, meaning that all pulses recorded for a range gate had about the same received power. In the lines marking the interference from the radar EIS, STD reaches up to 1, showing that the received power within all pulses varies strongly, hinting at a pulsed source transmitter.

### 3.2 Timeseries of radar-radar induced interferences

All dedicated radar MHP precipitation sweeps during the period of the experiment are analyzed for interference signals applying the previously introduced detection approach. For each sweep, the total number of pixels with an interference signal is stored with the timestamp of the acquired sweep. The result from each sweep is then correlated to the respective network radar that was pointing to the MHP radar. This is complemented with a reference precipitation scan without a planned interference. With this, a time series of interference numbers is obtained which will be analyzed in the following. Figure 5 shows how the number of pixels with an interference signal in MHP radar data changes during the experiment. Three distinct periods are visible. The first period spans from the beginning of the experiment to 2019-12-09, 15:30 UTC. During this time, MHP as the



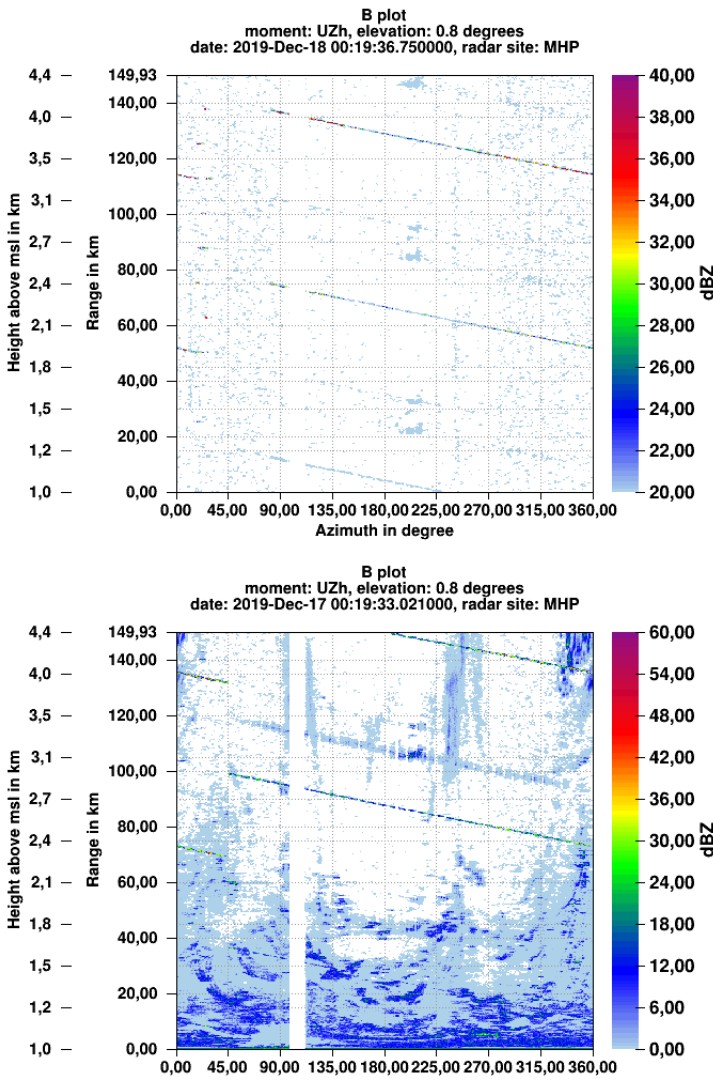

**Figure 3.** B-Plots of uncorrected reflectivity UZh illustrating the interference from the Eisberg (EIS) radar (lineshape structure). Upper Plot, transmitter turned off, MHP radar only in receive mode, bottom plot, MHP transmitter operating. Corresponding STD B-plots are shown in Figure 4.



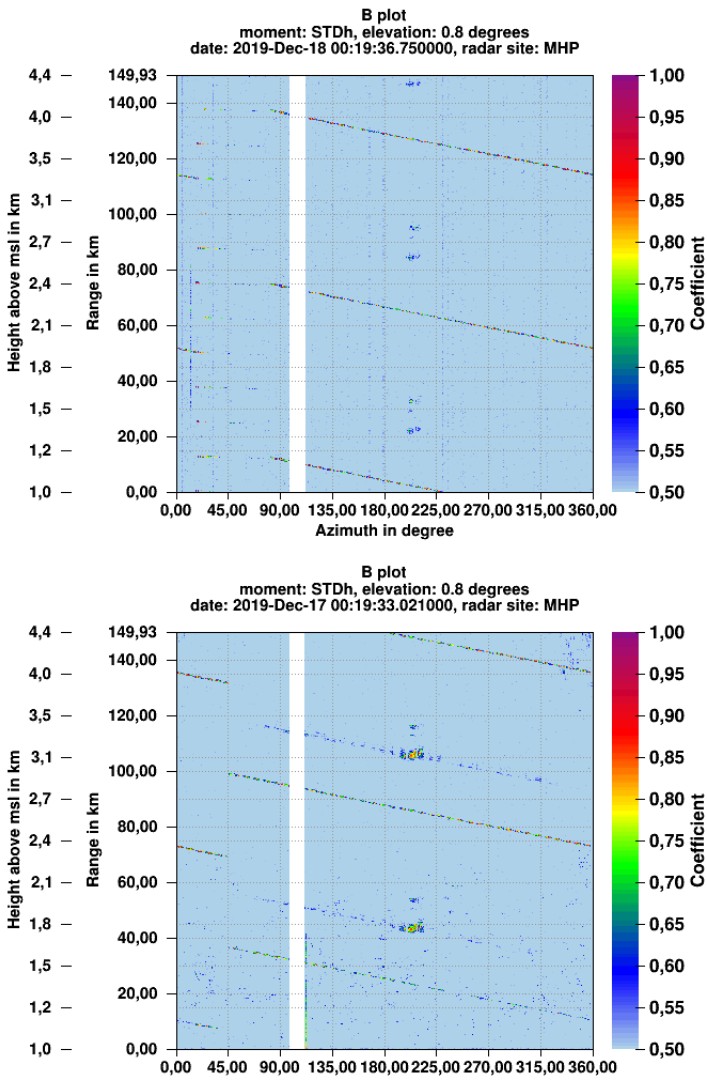

**Figure 4.** B-Plots of STDh illustrating the interference from the Eisberg (EIS) radar (lineshape structure). Upper Plot, transmitter turned off, MHP radar only in receive mode, bottom plot, MHP transmitter operating. At MHP, sector blanking is configured between 99 and 111° azimuth because of a church in the safety perimeter.





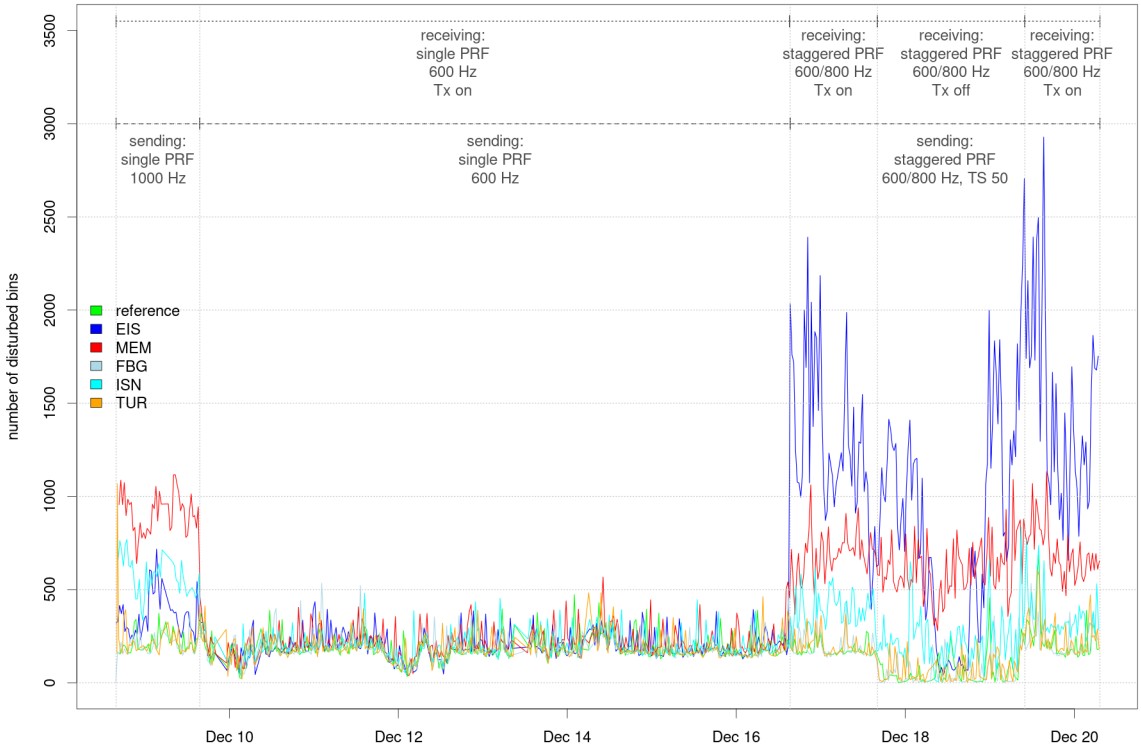

**Figure 5.** Time series of the number of pixels with an interference signal caused by the operational radars Eisberg (EIS), Memmingen (MEM), Feldberg (FBG), Isen (ISN) and Türkheim (TUR). Data from the radar Hohenpeißenberg (MHP) are evaluated to identify the inteference signal. The dotted line at the top shows the periods of different settings on the receiving MHP radar, the dashed line those of the transmitting radars.

receiving radar is operated at a single PRF of 600 Hz, while the fixed scan radars are set to a PRF of 1000 Hz. This difference in the settings increases the chance that interference signals are observed by MHP. The largest number of disturbed pixels is found when the radars MEM, ISN and EIS are transmitting in fixed scan mode towards MHP. These three radars are operated

within 15 MHz of the MHP transmit frequency (cf. Table 1). This provides a first upper limit of the frequency separation among magnetron radars for which interference is possible. With this configuration, the largest number of interference pixels are caused by the MEM radar, with ISN being second and EIS third. This may be attributed to a range dependence: MEM is closest with 65 km distance, EIS is furthest away with 219 km (also cf. Table 1). A statistical view on the results of the first period is shown with boxplots which provide a frequency distribution of the number of disturbed pixels (left), the correspond-

ing signal to noise ratio (SNR, center) and the cross-correlation coefficient (RhoHV, right, Figure 6). An increased number of interference pixels is found up to a frequency difference of <15 MHz (left, Figure 6)). This the case for the radars EIS, MEM and ISN. The largest number is found for the closest radar MEM. The smallest number (but still increased compared to the radars TUR and FBG) is found for the radar EIS. The number of MHP interference pixels in the presence of the radars TUR





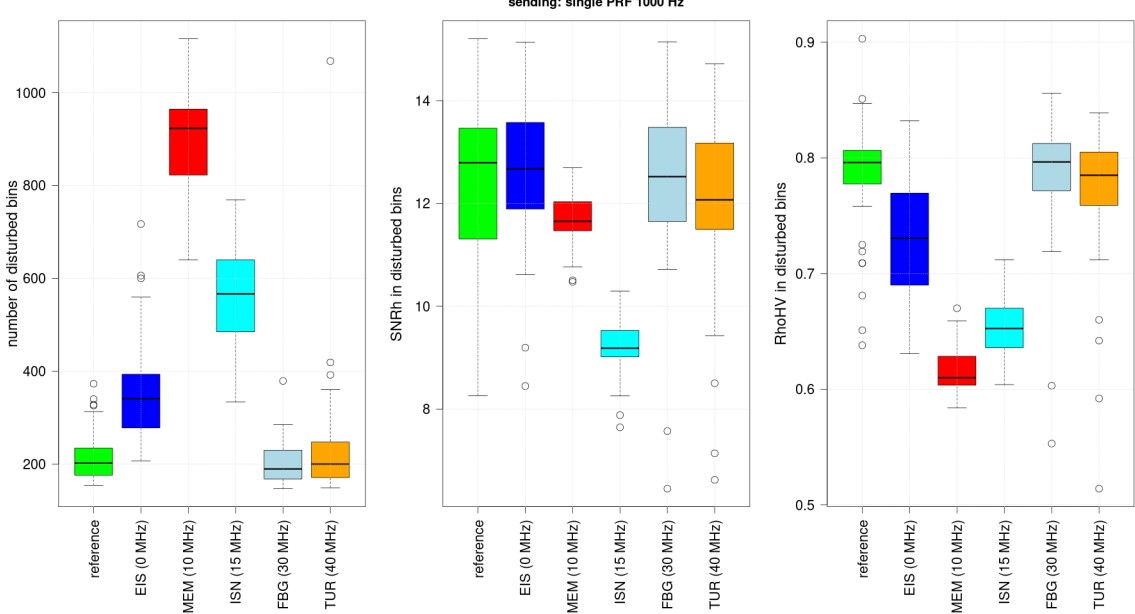

**Figure 6.** Boxplot of the number of disturbed pixels (left), the SNR in these pixels (center) and the RhoHV in these pixels (right) during the first period as described in 3.2. The values behind the radar names give their difference in sending frequency relative to the MHP radar.

and FBG is similar to the reference scenario where no dedicated source of interference is present. Since WiFi transmitters are

commonly operated at C-band, it is not surprising that there are always data with an interference signal, though the the total number of disturbed radar pixels is small in this case. Radar range bins with an interference have an SNR of about 13 dB. The ISN radar is causing a somewhat smaller SNR level (about 8 dB). Corresponding RhoHV values are between 0.6 and 0.8. This range of values is usually related to non-meteorlgical echos unless there is a presence of hail. That is obviously not the case considering the rather small SNR values.

The second period starts at the end of the first and ends at 2019-12-16, 15:00 UTC. During this time, both the receiving radar and the transmitting radars are operated at a single PRF of 600 Hz. This setup visibly minimizes the number of disturbed pixels. Variations during this period are caused by small changes in the elevation of the transmitting radars and by changes in weather, causing different conditions for propagation. The evaluation of SNR and RhoHV for this period (not shown) confirms these findings with no discernible differences among the radars.

During the third period, starting at 2019-12-16, 15:00 UTC and ending at the end of the experiment, both the transmitting and receiving radars are operated at a staggered PRF of 600 and 800 Hz. This clearly has a great effect on the number of disturbed pixels. Again, as in the first period, disturbances are caused only by the three radars operating within 15 MHz of the MHP radar (EIS, MEM, ISN). The largest number of interferences is caused by the EIS radar which is operating at the same frequency as the MHP radar. The MEM radar comes second, the ISN radar third. Within the third period the transmitter of

the MHP radar was turned off for two days (see Table 4). This time is clearly visible in Figure 5: the amount of pixels with




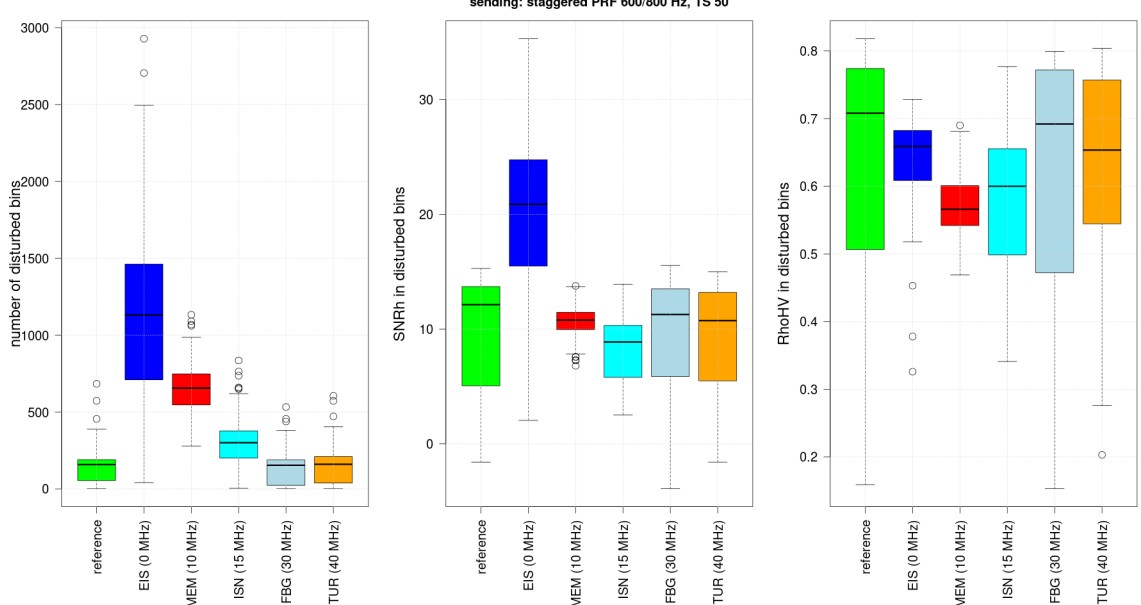

**Figure 7.** Boxplot of the number of disturbed pixels (left), the SNR in these pixels (center) and the RhoHV in these pixels (right) during the third period as described in 3.2. The values behind the radar names give their difference in sending frequency relative to the MHP radar.

an interference during the reference precipitation scan and the radars with a transmit frequency offset of more than 15 MHz (FBG, TUR) drops to almost zero. When the transmitter is turned off, only a small fraction of the back-scatter signal satisfies the interference criterion (SQI < 0.6, STD > 0.6, cf. Section 3) which we refer to as background interference noise. Also during this time there is a period in which the amount of disturbed pixels caused by EIS drops to comparatively low values. This

is presumably caused by a weather situation with unfavorable propagation conditions. Such conditions make the EIS radar (which is over 200 km away) invisible to the MHP radar. A similar drop is not present for the closest radar MEM when it transmits towards the radar MHP. Due to the proximity of this radar (65 km), unfavorable propagation conditions will not cause a significant decrease of detected interference pixels.

The corresponding boxplot for the whole third period is shown in Figure 7 which again illustrates that the number of

disturbed pixels is dependent on transmit frequency. Largest SNR values are measured when the transmit frequencies exactly match (EIS/MHP). Median SNR is then on the order of 20 dB. SNR reaches similar levels of about 14 dB for all other radars and the reference data set. The already mentioned frequency dependence is also revealed by the increased number of interference pixels within the 15 MHz band from the MHP radar. RhoHV values are between 0.6 and 0.8 which is indicative of a non-meteorolgical signal.





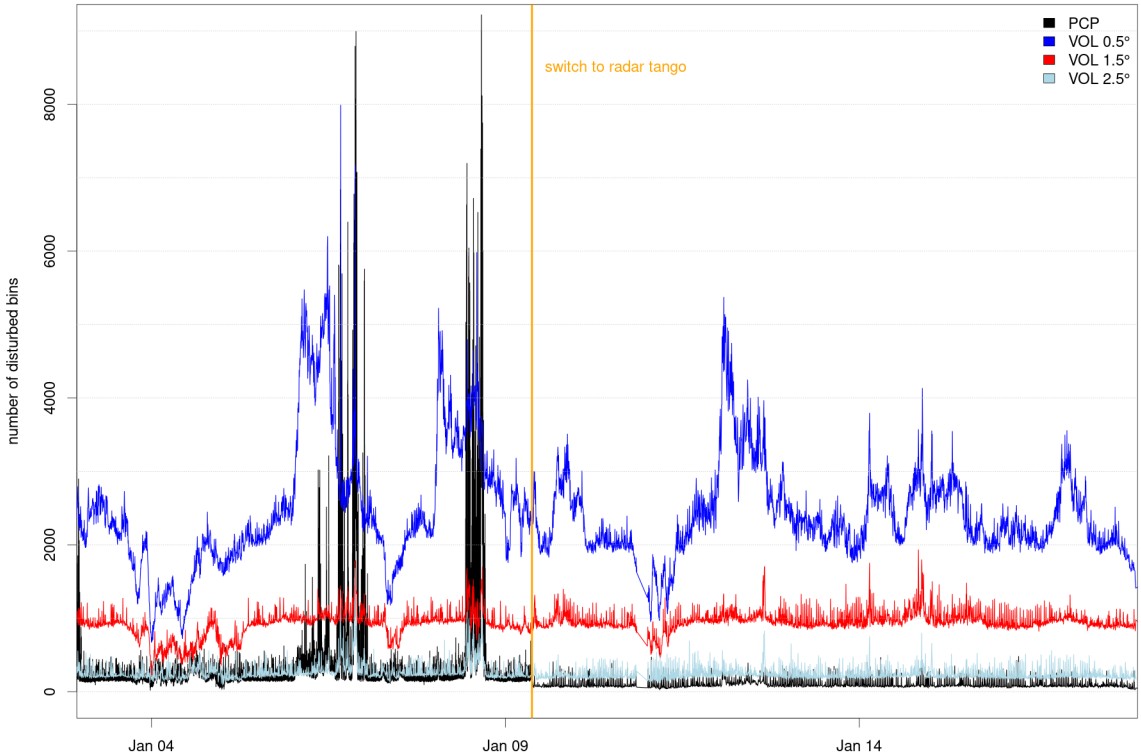

**Figure 8.** Time series of interference bins in the MHP PCP-scan and the lowest three elevations of the VOL-scan. The orange line marks the beginning to the radar tango.

## 3.3 Radar Tango

The effectiveness of synchronizing the radar scanning to minimize the radar-radar interference in the DWD radar network is investigated in this section. The synchronization of the radars is referred to as the "Radar Tango". At the beginning of every 5-minute scanning period, all 17 radars point to North (0° azimuth) and start the scanning simultaneously. Then they perform their "dance", with all radars approximately pointing towards the same azimuth angles at the same time. In doing this, it is always avoided that two antennas point towards each other at the same time. The "Radar Tango" has been introduced in Summer 2017. Up to now, there has not been a quantitative analysis on its effectiveness, but no obvious radar-radar induced interferences were reported by users since then. Intuitively, this is an expected result, but in order to obtain a quantitative view on this we set up a test sequence in January 2020. Beginning on 2020-01-03, we operated the MHP radar in an un-synchronized mode with the radar network. We then synchronized the scanning on 2020-01-09. For an overall assessment, we analyzed both the precipitation scan and the volume scan (the lowest three elevations 0.5, 1.5 and 2.5°). The effect of the radar tango is shown in Figure 8. Before the start of the "Radar Tango" on 2020-01-09, the amount of disturbed bins in the PCP scan irregularly increases by two orders of magnitude for a limited time. After the synchronization of the scanning, these spikes disappear. In



contrast, the three lowest elevations of the VOL-scan reveal no obvious change after the synchronization. Most interferences are found in the lowest elevation of 0.5°. There appears to be a correlation to the increased PCP interference numbers before

the synchronization. However, after the synchronization, the temporal variability and the overall number of rangebins with an interference signal is similar to the initial un-synchronized period.

One reason for the relatively large interference numbers in the lower elevations of the VOL-scan is attributed to the surrounding orography: when the radar is pointing towards the Alpine ridge towards south at low elevation, the chance of detecting an external microwave source is increased. The elevation adjustments of the PCP to avoid blocking from the orography, effectively

avoids sources of interference (Figure 8). An increase of already 1° in elevation substantially decreases the number of range bins with an interference signal.

### 3.4 Discussion

Based on the previous analysis, radars separated by more than 15 MHz in transmit frequency from the MHP radar cause no interference signal. In addition, the "Radar Tango" proves to be an effective method to avoid radar-radar interference. The

plausibility of this is now discussed by considering the effect of the transmitter and receiver components on the characteristic spectrum of a magnetron pulse with a given bandwidth. The following is illustrated with Figure 10.

In order to suppress spurious emissions, which are not allowed by regulations, a bandpass (200 MHz bandwidth) and a harmonic filter (7 GHz lowpass) are built into the transmit waveguide before splitting the pulse in horizontal and vertical polarization. With an additional filter (lowpass 6 GHz) in both H & V, installed behind the waveguide splitter, side emissions

are suppressed below the required thresholds. In the analog part of the receiver, the received signal passes through a pre-select lowpass filter which has a bandwidth of 22 MHz. The signal, which at this stage is still at radio frequency, is mixed to IF (intermediate frequency) at 60 MHz using the actual TX frequency of the burst measured with reference to the stable local oscillator (STALO). After the digitization of the received signal in the so called intermediate frequency digitizer (IFD), a matched filter is applied to improve the signal to noise ratio. This is done pulse by pulse. The bandwidth of the matched filter

is about inversely proportional to the pulse length. For the 0.8 μs pulse width, the 3 dB bandwidth is 1.4 MHz. The IFD output is I&Q data which is processed in the signal processor to compute radar moments. We can now raise the question, why a radar system with a bandwidth of 1.4 MHz that is operating at a 15 MHz difference from the nominal transmit frequency of the MHP radar, can still disturb the MHP data. To illustrate this, we make a simple back-of-the-envelope calculation adapting an ITU recommendation (ITU, 1997). The power $P_{rx}$ received by the MHP radar can be estimated as:

$$P_{rx} = P_{tx}@f_{rx} - FSPL + G_{tx} + G_{rx} - A_G - L_{tx} - L_{rx} \tag{3}$$

with FSPL being the free-space path loss, $A_G$ the gas attenuation, $L_{tx}$ and $L_{rx}$ the transmit and receive losses, respectively, and the antenna gain of the receiving and transmitting antenna, $G_{rx}$ and $G_{tx}$. The term $P_{tx}@f_{rx}$ refers to the transmitted power of the magnetron in the nominal operating frequency band of the receiving radar. The FSPL (in dB) can be written as

$$FSPL = 10\log_{10}\left(\frac{4\pi d}{\lambda}\right)^2 \tag{4}$$





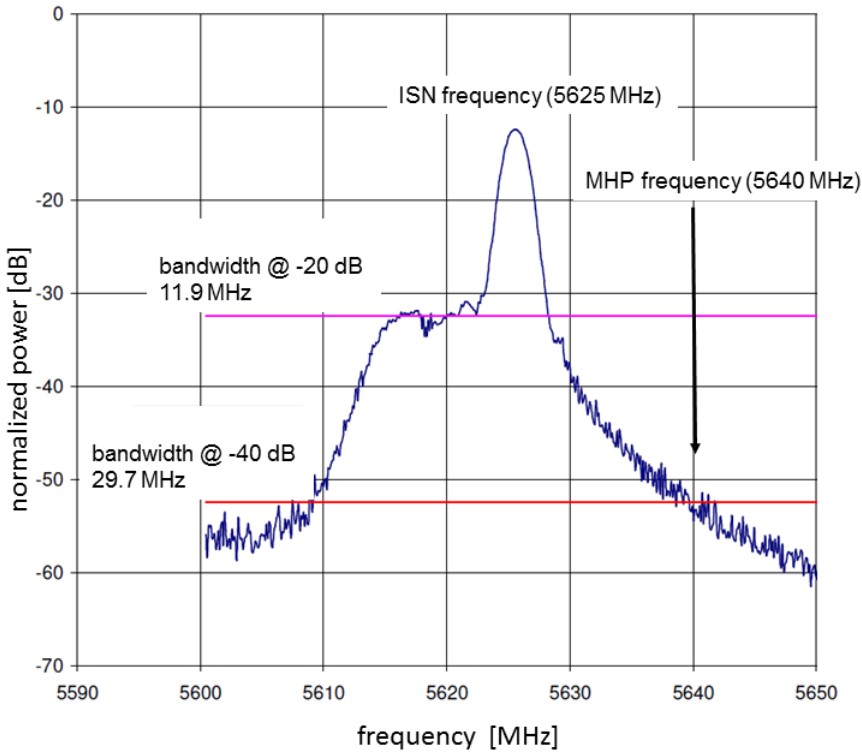

**Figure 9.** Measured frequency spectrum of the ISN magnetron for a pulse length of 0.4 μs. Measurements were taken between magnetron and the wave guide filters. The resulting bandwidths at -20 dB and -40 dB are shown. The ISN transmitter is operated at 5625 MHz. For illustrative purposes, the MHP transmitter frequency (5640 MHz) is shown as well.

with the radar wavelength $\lambda$ and the distance $d$ between the two radars.

We consider the radar ISN, which operates at 5625 MHz and is 91 km away from the radar Hohenpeißenberg. We now want to compute the expected power at the MHP low-noise amplifier (LNA) using the measured emitted power of the ISN magnetron at the MHP radar frequency 5640 MHz. A measured spectrum of a 0.4 μs pulse is shown in Figure 9. For a pulse length of 0.8 μs the 3 dB bandwidth is smaller, but the overall shape is similar. The measured bandwidth of the ISN pulse at

the 40 dB level (40 dB below the peak) is about 30 MHz. Even though the pulse is not strictly symmetric, the power spectral density at 5640 MHz is about 40 dB below the peak at 5625 MHz. The total transmit power is about 86.5 dBm and is reduced by approximately 2 dB of transmit losses and another 3dB for the power splitter. The major part of the power is concentrated in the desired 1.4 MHz band around 5625 MHz. In the same bandwidth at 5640 MHz, a total power of roughly $86.5 - 2 - 40$ dBm can be estimated. Antenna gain on the transmit and receive path is 45 dB. With a receive loss of 3 dB and the given distance of

91.5 km, which results in a FSPL of 146 dB, we can approximate about -18 dBm of power from the ISN radar in the matched





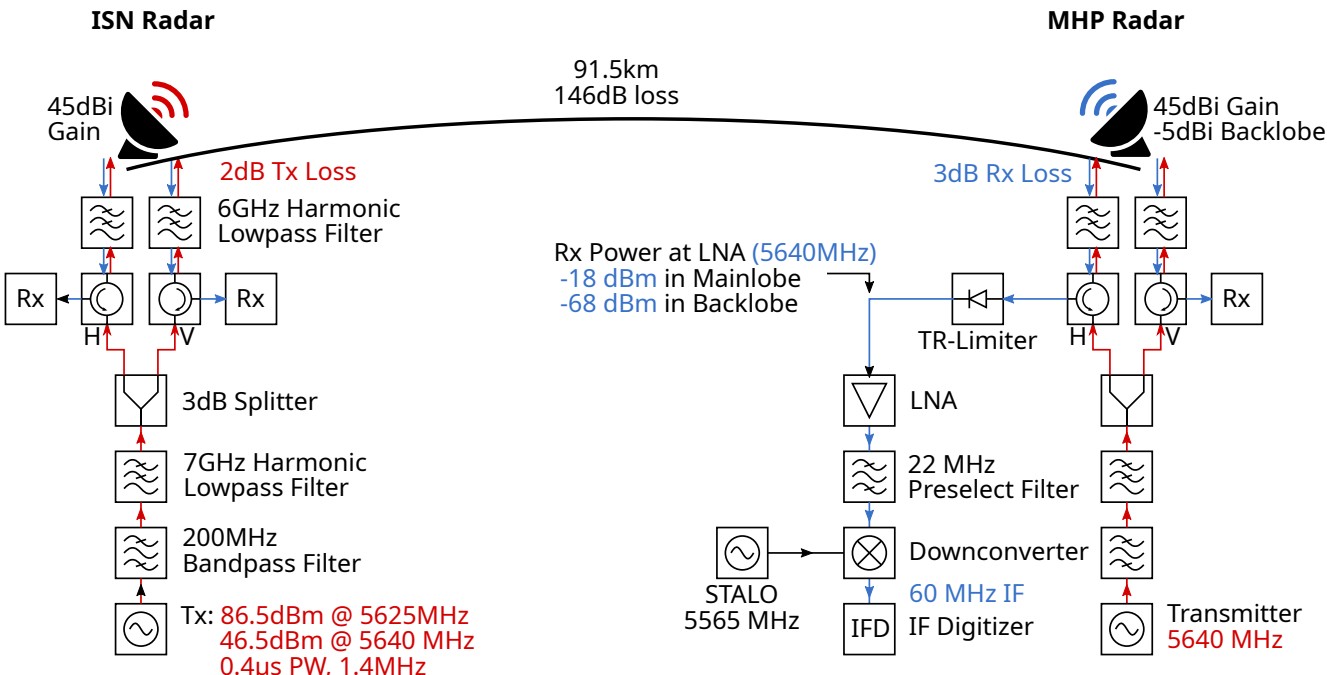

**Figure 10.** Radar setup used for the analysis. The transmit signal from the ISN radar at 5640 MHz interferes with the Rx at MHP radar at 5625 MHz. Important RF components are included as reference

filter bandwidth of the MHP radar in the LNA reference plane. This would result in a relatively strong signal which is well within the linear sensitivity range of the receiver (-110 to -7 dBm). The results represent a worst case scenario, since the two radars do not have a free line of sight and the total signal loss on the transmission path will be significantly higher in reality. Gas attenuation has a negligible effect amounting to 0.7 dB. Note that it is not necessary to consider the waveguide filters

because their relatively large bandwidth does not significantly alter the analyzed part of the pulse spectrum.

  For the Radar Tango, a similar estimate can be made by assuming a gain of -5 dB (backlobe of the antenna) on the receiving, and a gain of 45 dB for the transmitting system. The other specifications remain the same. It is important to realize that this is the case for just one azimuth angle when a radar pair is perfectly aligned with free line of sight. For the precipitation scan (12 °s$^{-1}$ antenna speed) this occurs for about 0.08 s. For this special configuration, a power of -66 dBm could be expected at

the MHP LNA. According to the antenna diagram (see Frech et al. 2013), the expected power drops below -100 dBm within about 3°. This is because antenna sidelobes are below 40 to 50 dB beyond 3°. Outside the perfect radar pair alignment we then can assume a gain of -5 dB for both antennas. For this simplified scenario we could expect -118 dBm at the LNA of the MHP radar, which is well below the sensitivity level.

  This simple back-of-the-envelope calculation illustrates that a combination of radar siting, orography and frequency spacing

helps to minimize radar interference. The most effective mitigation of interference however can be achieved by the Radar



Tango. Except when there is a perfect alignment of the radars, the expected power at the LNA is below the radar sensitivity. Interferences in the situation of a perfect alignment in azimuth is not seen in day-to-day operations. This can be attributed to the fact that all radars scan at an elevation $\geq 0.8°$, so that only the power from sidelobes may play a role. Then the corresponding power levels are small (see above), so that there is no interference even during the brief instances of perfect alignment. This is supported by the measurements in this study and from the day-to-day experience after the introduction of the Radar Tango in the DWD radar network.

## 4 Summary

The goal of this paper was to assess whether additional C-band radars can be added to the DWD radar network, given the available 50 MHz bandwidth in this frequency band. In order to derive a practical guideline, we have set up an experiment where five operational radars at different transmit frequencies serve as possible sources for interference. The reference radar is the research radar Hohenpeißenberg, which operates at a frequency of 5640 MHz. In order to identify an interference, we employ a combination of two radar data quality moments (STD and SQI), which are successfully used to identify and eliminate external interference sources in the German weather radar network (Schaper et al., 2022). An important goal of the experiment was to consider a range of meteorological situations which are common in normal day-to-day operations. We also assessed the effectiveness of the "Radar Tango" which has been introduced to minimize the radar-radar induced interference in Summer 2017. Based on the results, a number of conclusions can be drawn:

- as an obvious result, interference is strongest when both the transmitting and the receiving radar operate at the same transmit frequency.

- The interference signal from an external source is only seen up to a transmit frequency separation of about 15 MHz. In our case, this result is found for the radar ISN which operates at 5625 MHz. This result may vary for a different magnetron transmitter/filter setup.

- For identical frequencies, SNR is on average near 20 dB. For all other frequencies, SNR is about 14 dB, while there is a strong frequency dependence with respect to the number of range bins with an interference signal. Time series of interference pixel numbers show a lot of variability in the number of range bins with interference which relates to variability in propagation effects and, to a minor extent, the random sampling of the interference by the MHP radar.

- when allocating a transmit frequency, radars that are over 200 km away have to be included in the considerations. This is illustrated by the clear interference signal in the MHP radar data caused by the radar EIS at 219 km distance.

- the Radar Tango is the most effective means to minimize the radar-radar interference (up to 50 dB higher suppression). Time periods with an increased number of disturbed pixels are found when the radars are not synchronized every 5 minutes to start with the same azimuth angle.




– the relatively broad frequency spectrum of a typical magnetron pulse may cause interference in a large frequency range within the usable C-band. This issue can be efficiently mitigated by the Radar Tango.

To summarize, these results provide a guidance to determine a proper frequency assignment in a C-band weather radar network such that the possibility of a radar-radar induced interference is minimized. When selecting the appropriate frequencies,

even radars at a distance around 200 km have to be considered since they have a chance to become visible to a radar (in our case the EIS-MHP radar pair) in stably stratified situations with favorable refractive gradients. An increased number of interferences is found up to a frequency separation 15 MHz. Beyond this limit, the number of range bins with an interference signal is comparable to data where no dedicated external interference signal is present.

Most importantly, the possibility of a radar-radar induced interference is minimized if the scanning of the radars is synchro-

nized. These results suggest that additional C-band systems can be added to the German weather radar network with 17 radars without causing additional radar-radar induced interference.

*Data availability.* Data sets used for this study are availabe upon request.

## Appendix A: Scan parameterizations

The parameterizations of the operational DWD precipitation (PCP) and volume (VOL) scans used during this investigation are

shown in Table A1, Table A2 and A3, respectively. Note that the range bin resolution in the volume scan has been increased in spring 2021 from 1 km to 250 m.

**Table A1.** The scan parameterization of the operational Precipitation-Scan (PCP)

| Parameter | value |
|---|---|
| Elevation [°] | variable (terrain following) |
| PRF [Hz] | 600 |
| Pulse width [µs] | 0.8 |
| Azimuth speed [°s$^{-1}$] | 12 |
| Raw range bin size [m] | 25 |
| Range resolution [m] | 250 |
| Range [km] | 150 |
| Beam width [°] | 1 |



**Table A2.** Elevation angles as a function of azimuth angle of the operational PCP-Scan at MHP.

| Azimuth [°] | Elevation [°] |
|---|---|
| 270 - 80 | 0.8 |
| 80 - 125 | 2.0 |
| 125 - 160 | 3.0 |
| 160 - 195 | 3.5 |
| 195 - 220 | 3.0 |
| 220 - 240 | 2.5 |
| 240 - 270 | 2.0 |

**Table A3.** Scanning parameters of the operational Volume-Scan (VOL). Two values for PRF implies staggering.

| Parameter | sweep0 | sweep1 | sweep2 | sweep3 | sweep4 | sweep5 | sweep6 | sweep7 | sweep8 | sweep9 |
|---|---|---|---|---|---|---|---|---|---|---|
| Elevation [°] | 5.5 | 4.5 | 3.5 | 2.5 | 1.5 | 0.5 | 8 | 12 | 17 | 25 |
| PRF [Hz] | 600/800 | 600/800 | 600/800 | 600/800 | 600/800 | 600/800 | 800/1200 | 2410 | 2410 | 2410 |
| Pulse width [µs] | 0.8 | 0.8 | 0.8 | 0.8 | 0.8 | 0.8 | 0.8 | 0.4 | 0.4 | 0.4 |
| Azimuth speed [°s$^{-1}$] | 16 | 16 | 16 | 16 | 16 | 12 | 18 | 30 | 30 | 30 |
| Raw range bin size [m] | 25 | 25 | 25 | 25 | 25 | 25 | 25 | 25 | 25 | 25 |
| Range resolution [m] | 1000 | 1000 | 1000 | 1000 | 1000 | 1000 | 1000 | 1000 | 1000 | 1000 |
| Range [km] | 180 | 180 | 180 | 180 | 180 | 180 | 124 | 60 | 60 | 60 |
| Beam width [°] | 1 | 1 | 1 | 1 | 1 | 1 | 1 | 1 | 1 | 1 |

*Author contributions.* MF designed the experiment, carried out the measurements and parts of the analysis. Main portions of the article were written by MF. CH analyzed the data and wrote parts of the article. MS implemented the interference detection, ML contributed to the technical description of the radar system, BR and BL contributed to the analysis and interpretation of the results.

*Competing interests.* There are no competing interests

*Acknowledgements.* The fruitful discussions with the DWD radar development team are gratefully acknowledged. We want to acknowledge in particular the significant contribution of Theo Mammen and Kay Desler. Their exceptional work formed the basis for this study.



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
