# Peer review of "Assessing and mitigating the radar - radar interference in the German C-band weather radar network"

_EGUsphere, 2022_

## Author Response (AR1)

**Reviewer 1 comment**: "the definition of STD is not precise and seems to be a special product of the GAMIC signal processor. Also, the Schaper et al. 2022 preprint does not give sufficient details. In line 56 STD is defined as the standard deviation of the power using I&Q data. In Schaper et al. 2022 it is defined as a normalized standard deviation of received power. Obviously, the expected range of 0 .. 1 implies normalization. But to which value? What is "mean" STD (line 62) in this context? The authors should give more details for users not familiar with the GAMIC signal processor. I also can not follow the statement that STD app. 0.5 indicated all pulses have the same power (line 156). If in a timeseries all data have the same value, standard deviation is 0! SQI has a more common definition, it is provided by a number of signal processors, and is generally defined as the magnitude of autocorrelation lag 1 divided by autocorrelation lag 0; or simply 1 by (spectral width normalized with Nyquist interval). Assuming that Tx phase is removed from measured phase angle."

**Response**:  STD: We operate the signal processor Enigma in a DAS ("dynamic angle syncing" mode). Radar moments are computed from the batch of received pulses that are collected for the predefined angle width (1°). The number of processed pulses depends on the antenna speed and the PRF. For a batch of pules, STD is the normalized standard deviation of the I/Q magnitude in a given range bin. The standard deviation of the I/Q power in the batch is normalized with the mean I/Q power. Pulsed interferences usually are not correlated with the radar PRF, and the standard deviation of the I/Q power magnitude is increased because the interference is expected to be not present in every single pulse. The noisiness then increases, and the SQI decreases. This is the principle idea behind the usage of STD and SQI for the interference detection.

To clarify the description, we will change "standard deviation of the power" in line 56 to "normalized standard deviation", remove the word "mean" in line 62 and add some more in depth explanation of STD in the revised version. Line 156 will be rewritten to: "In most of the scanned area, STD is at ~0.5, indicating a typical value for recorded noise. A stable weather signal usually is at STD < 0.2.".

**Reviewer comment**: the authors should give some more explanation why the interferences are reduced in the case where all radars operate with the same PRF (line 187).

**Response**:  Thanks for pointing this out. We have  included the following section:

Unless the transmitted pulses are exactly synchronized in time, there is only a small chance to capture the transmitted pulse from one of the external radars of this experiment. However, the pulse trigger timing cannot be synchronized because the trigger timing of a particular scan depends on the timing of the previous scans which involves for example the time it takes to position the antenna. This will be variable from system to system. Another aspect to keep in mind is that the receiving PCP scan only samples data up to a range of 150 km, whereas the unambiguous range is 250 km. So any pulse which would appear in a range beyond 150 km cannot be captured on the receive side. However with different PRF settings, there always will be pulses that are captured by the MHP system within the scan time of 30 seconds.

Line 30: is 2 km in Fig.1

-Thank you. We will change the number from 3 to 2 in the revised manuscript

Line 41: explain PRF

-PRF (pulse repetition frequency) is already explained in line 21/22.

Line 110: receive end: MHP? Transmit end: other radars?

-Correct. We will add the short names of the respective transmitting and receiving radars in brackets.

Line 125: $H_b$ above radar?

-$H_b$ in this case describes the beam height over an imaginary "earth" with a starting radius of $r_e$ (which depends on latitude, elevation and finally the radars height over ground) and the 4/3 factor. So we think "above ground" is more fitting than "above radar".

Line 154+155: better: The received power is independent from range.

-Agreed. We will change the text to your suggestion.

Line 229: refer to Table A2 for the elevations used by PCP at MHP

-Good suggestion! We will add a cross reference to the table in the revised manuscript.

Figure 10: to what does 1.4 MHz refer? Red numbers at lower left side

-The number of 0.4 µs was wrong in the figure and will be replaced by 0.8 µs in the revised version. The 1.4 MHz refers to the 3dB bandwidth of the transmitted pulse with a pulse duration of 0.8 µs.

Line 283: in Table A3 sweep 5 is at 0.5° elevation

-We will change the number in the sentence to 0.5°. For some additional info on the chosen elevations:

Sweep number 5 is indeed at 0.5° elevation. The elevation steps of 1° are chosen according to the antenna beam width (roughly 1°) at low elevation (up to 5.5°) in order to obtain a good volume sampling close to the surface. Compared to the precipitation scan, a 0.5° elevation scan may contain more clutter, but at the same time also contains valuable radar information close to the surface. The 0.8° elevation of the precipitation scan has been chosen empirically in order to provide good quantitative precipitation estimates close to the surface and to avoid ground clutter.

**Reviewer 2**

**Reviewer 2 comment:** RFIs at C-band are nowadays widespread in Europe. The study focuses on RFIs caused by other weather radars. In the data analysis, it is not clear how, in the data analysis, wifi RFIs are distinguished from weather radar ones. The amount of wifi RFIs can be partially inferred by Figure 6, where about 200 bins are classified as disturbed.

**Response**: We did not try to eliminate the WIFI disturbances. We analyze the relative difference in the numbers of the identified classified rangebins. Any differences are then attributed to the respective radar configuration of the transmitting radar. This makes the assumption, that the WIFI disturbance is constant throughout the considered time period. For the MHP radar this is a valid assumption based on the continuous monitoring for WIFI interferences as described in Schaper at al. (2022).

**Reviewer comment:** A short description of current meteorological conditions, focusing on atmospheric refractivity, could help to better interpret the analysis. Moreover, it is unclear which PW is set up in the experimental scans.

**Response** First, all transmitting radars were using a PW of 0.8 µs during the experiment. We will add this information to table 3. The PWs of the receiving radar MHP is described in the appendix, tables A.1 and A.3. All evaluated sweeps use a PW of 0.8 µs as well.

Second, we agree that showing the connection between disturbed bins and refractivity gradients would be helpful. We will add a new paragraph at the end of section 2 where we calculated refractivity gradients from radiosonde data and compare the frequency of occurrence of strongly negative values with the amount of recorded interferences. This will be supported by an additional figure. It will show that we have more disturbed bins when the radiosondes show lower values for the refractivity gradient.

**Reviewer:** Figure 6 and Figure 7 summarize the number and properties of disturbed bins:  RhoHV shows a wide variability: it's worthing to add some comments trying to explain possible causes.

**Response:** First of all, RhoHV is the cross correlation coefficient from time series where no Doppler-Clutter filter has been applied. There is no further processing or quality control (e.g. thresholding; this information will be in the revised manuscript) applied. The large variability in RhoHV reflects the typical observed value range if there is no precipitation, and where the backscattered signal is caused by targets (e.g. insects, moving ground clutter) that are often without correlated backscatter in the two polarization planes.

The uncorrected RhoHV also has a strong dependence on SNR: if SNR is below 0, RhoHV never reaches values above 0.6 even in pure rain. at SNR ~10 dB, RhoHV can take almost all values between 0 and 1 and only above that RhoHV takes on stable values. RhoHV in figure 6 therefore has a small variability, since most SNR values lie stably above 10 dB. In figure 7, variability in RhoHV is higher, because more bins have a SNR clearly below 10 dB which results in a lower RhoHV. We will add a few new sentences at the end of section 3.2 that explain this connection.

Also please note that with your suggestion of having matching axes in both figures it becomes

very clear that variability of RhoHV is only high in figure 7. The boxes in Figure 6 for RhoHV become very small when showing the whole range from 0 to 1.

**Reviewer**: The German radar network shows large overlapping areas where 2D or even 3D winds can be retrieved. Do the authors believe that the "Radar Tango" scan strategy affects or limits such retrieval? The reviewer suggests a discussion on the possible drawbacks of "Radar Tango".

**Response:** We do not expect a disadvantage. This is supported by the evaluation of the dual-Doppler retrieval of the 3-D wind vector from volume data with multiple sweeps we just did for another study. Even without the Radar-Tango, there always will be time differences in sampling a specific volume from 2 or more radars. So a stationarity assumption always has to be made, which is, depending on the meteorological phenomena, more or less an approximation, especially if data from multiple sweeps are used.

**Minor remarks of the reviewer:**

Figure 1 - Add scale bar and North Arrow, please.

- We will add a scale bar to the figure. A North Arrow would not be useful in our opinion. The projection of the map is such that the longitudes converge towards north. If we put the arrow in the lower left corner, for example, it would only be valid for this exact longitude. The north direction in the center or on the right hand side of the map is a different one. To clarify the position and direction of the map we will add the following sentence to the description: "The grid is in latitudes north and longitudes east."

Figure 2 - Add scale bar and North Arrow, please. Adding distance in km at each arrow could improve clearness.

-We will add all requested items to the map in the revised version.

Figure 6 - change "Boxplot of the number of disturbed pixels" to "Boxplot of the number of disturbed bins", please

-We will change the annotations from "pixels" to "bins" in Figures 6 and 7.

Figure 6 and Figure 7 should report the same range on both axes.

-We agree, the difference is more clearly seen with matching ranges. We will adopt your suggestion in the revised version of the manuscript.